# Inhibition of Spinal TRPV1 Reduces NMDA Receptor 2B Phosphorylation and Produces Anti-Nociceptive Effects in Mice with Inflammatory Pain

**DOI:** 10.3390/ijms222011177

**Published:** 2021-10-16

**Authors:** Suk-Yun Kang, Su Yeon Seo, Se Kyun Bang, Seong Jin Cho, Kwang-Ho Choi, Yeonhee Ryu

**Affiliations:** KM Science Research Division, Korea Institute of Oriental Medicine, Daejeon 34054, Korea; sy8974@kiom.re.kr (S.-Y.K.); ssy1025@kiom.re.kr (S.Y.S.); sichosi@kiom.re.kr (S.K.B.); ipcng@kiom.re.kr (S.J.C.); ddoongho@kiom.re.kr (K.-H.C.)

**Keywords:** transient receptor potential vanilloid 1 (TRPV1), *N*-methyl-D-aspartate (NMDA), glutamate, capsazepine, inflammatory pain

## Abstract

Transient receptor potential vanilloid 1 (TRPV1) has been implicated in peripheral inflammation and is a mediator of the inflammatory response to various noxious stimuli. However, the interaction between TRPV1 and *N*-methyl-D-aspartate (NMDA) receptors in the regulation of inflammatory pain remains poorly understood. This study aimed to investigate the analgesic effects of intrathecal administration of capsazepine, a TRPV1 antagonist, on carrageenan-induced inflammatory pain in mice and to identify its interactions with NMDA receptors. Inflammatory pain was induced by intraplantar injection of 2% carrageenan in male ICR mice. To investigate the analgesic effects of capsazepine, pain-related behaviors were evaluated using von Frey filaments and a thermal stimulator placed on the hind paw. TRPV1 expression and NMDA receptor phosphorylation in the spinal cord and glutamate concentration in the spinal cord and serum were measured. Intrathecal treatment with capsazepine significantly attenuated carrageenan-induced mechanical allodynia and thermal hyperalgesia. Moreover, carrageenan-enhanced glutamate and phosphorylation of NMDA receptor subunit 2B in the spinal cord were suppressed by capsazepine administration. These results indicate that TRPV1 and NMDA receptors in the spinal cord are associated with inflammatory pain transmission, and inhibition of TRPV1 may reduce inflammatory pain via NMDA receptors.

## 1. Introduction

The transient receptor potential (TRP) family is one of the most important and well-studied groups of ion channels involved in biological processes. TRP channels are a group of ion channels located predominantly on the plasma membrane of cells. These channels mediate a variety of sensations such as temperature, touch, osmolarity, and chemicals that cause painful sensations [1]. Activation of TRP channels enables crosstalk between neurons and immune cells to regulate inflammatory processes and permits detection of noxious stimuli by triggering action potentials in peripheral nociceptors [2,3]. TRP vanilloid type 1 (TRPV1), a TRP channel known as the capsaicin receptor, is widely expressed in peripheral and central nervous system regions involved in pain transmission and modulation. TRPV1 plays a major role in sensing noxious heat stimuli and mediating thermal hyperalgesia in inflammation [4]. TRPV1 is activated by inflammatory mediators such as protease, ATP, and bradykinin under inflammatory pain conditions and is upregulated in neuropathic pain conditions [5,6]. Inhibition of TRPV1 reduces arthritis severity and depletes neuropeptide levels induced by increased intracellular calcium [7]. Further, swelling of the knee joint and thermal hyperalgesia in TRPV1 knockout (KO) mice were reported to be reduced in CFA-induced arthritis [8]. Due to its role as a regulator of inflammatory pain, TRPV1 is a promising target for therapeutic intervention in the management of inflammatory pain.

Increased activity of the N-methyl-D-aspartate (NMDA) receptor in the spinal cord plays a critical role in the induction of central sensitization, an important process in the development and maintenance of pain hypersensitivity [9]. The functional regulation of NMDA receptors is achieved via NMDA receptor phosphorylation [10], which also modulates the increase in NMDA receptor activity in pathological states [11]. The NMDA receptors are composed of four subunits derived from the related families of NR1, NR2, and NR3, the typical NMDA receptors consist of two NR1 subunits that bind glycine and two NR2 subunits that bind glutamate [12]. The NR1 is an essential subunit that combines with NR2 or NR3 subunits to form a functional receptor [13], and has been implicated in inflammatory pain sensitization during inflammation-induced nociception [14]. We recently demonstrated that electrical stimulation suppressed the expression of phosphorylated NR2B in the spinal cord in chemotherapy-induced peripheral neuropathic pain [15,16]. The intracellular domains of NMDA receptor subunits contain consensus phosphorylation sites that regulate NMDA receptors for serine and threonine kinases. The most actively studied areas in terms of NMDA receptor regulation are protein kinase A (PKA) and protein kinase C (PKC). The calcium/calmodulin-dependent protein kinase II (CAMKII) is also known to translocate to NMDARs in an activity-dependent manner [17]. The activation of TRPV1 increases Ca^2+^ influx into neurons, followed by the activation of CaMKII, PKA, and PKC [18]. Collectively, these findings suggest that NMDA receptors contribute to inflammatory pain by modulating Ca^2+^ influx into neurons via TRPV1.

In this study, we investigated the effects of intrathecal administration of the TRPV1 antagonist, capsazepine, on carrageenan-induced inflammatory pain responses, such as mechanical allodynia and thermal hyperalgesia, in mice. We also evaluated whether the anti-nociceptive effects of capsazepine were related to the expression of phosphorylated NMDA receptors in the spinal cord or glutamate levels in the spinal cord and serum.

## 2. Results

### 2.1. Development of Nociceptive Behavior Using Carrageenan

Intraplantar injection of carrageenan produced significant mechanical allodynia and thermal hyperalgesia, as shown in Figure 1. In the ipsilateral hind paws, the carrageenan injection group (2% CR) exhibited increased paw withdrawal frequency (%) compared to control animals (Control) from 1 to 24 h after carrageenan administration (*** *p* < 0.001), whereas no significant difference was observed in paw withdrawal frequency (%) between saline-treated animals (Vehicle) and control animals from 1 h after carrageenan administration to the final behavioral measurement at 24 h (Figure 1A). No significant differences were observed in paw withdrawal latency (s) to thermal stimuli between the saline-treated group (Vehicle) and control animals (Control) at all points of behavioral measurement. Paw withdrawal latency of the ipsilateral hind paw in carrageenan-treated animals (2% CR) was significantly decreased from 1 to 24 h compared to that in control animals (** *p* < 0.01 and *** *p* < 0.001, Figure 1B).

### 2.2. Dose-Dependency of Capsazepine in Capsaicin-Induced Spontaneous Pain Test

To determine the dose underpinning the antinociceptive effects of capsazepine, we conducted a capsaicin test. The TRPV1 channel activator capsaicin induced paw-licking behavior characteristic of nociception, as shown in Figure 2. Intraplantar injection of capsaicin induced paw-licking time of 129.5 ± 37.0 s, which was significantly higher than that in the vehicle group (33.8 ± 28.2 s, *** *p* < 0.001). When capsazepine (0.1 μg) was administered via i.t. injections, capsaicin-induced paw-licking time was 118.5 ± 25.7 s, and no significant difference was observed compared to the capsaicin injection group. Doses of 1 and 10 μg resulted in a reduction in paw-licking time by 69.5 ± 19.3 and 66.5 ± 33.9 s, respectively, as compared with the paw-licking time of mice that received intraplantar injection of capsaicin (** *p* < 0.01). Intrathecal administration of sterile saline (vehicle) did not significantly reduce capsaicin-induced paw-licking time (112.3 ± 25.4 s).

### 2.3. Anti-Nociceptive Effects of Intrathecal Capsazepine Administration

The capsaicin test identified 1 μg of capsazepine as the appropriate dose to produce analgesic effects. Intrathecal administration of 1 μg of capsazepine resulted in transient antinociceptive effects on mechanical allodynia and thermal hyperalgesia after carrageenan injection compared to that in carrageenan-injected animals. Capsazepine-treated animals (CZP 1 μg [i.t.]) exhibited significantly lower paw withdrawal frequency (%) after carrageenan injection at 1 and 2 h after capsazepine administration compared to that in mice with carrageenan-induced inflammatory pain (2% CR, ** *p* < 0.01). In contrast, no significant differences were observed in paw withdrawal frequency (%) between saline-treated animals (Vehicle [i.t.]) and control animals (Control, Figure 3A). No significant differences were observed in paw withdrawal latency (sec) to thermal stimuli between the vehicle group (Vehicle [i.t.]) and control animals (Cormal) at all time points of measurement, but capsazepine-treated animals (CZP 1 μg [i.t.]) exhibited anti-hyperalgesic effects at 1 and 2 h compared to mice with carrageenan-induced inflammatory pain (Control, * *p* < 0.05, ** *p* < 0.01, Figure 3B).

### 2.4. Concentration of Glutamate in the Spinal Cord and Serum

Measurements of glutamate concentration confirmed the suppressive effect of intrathecal capsazepine treatment in the spinal cord and serum 1 h after carrageenan injection. Spinal levels of glutamate were significantly higher in carrageenan-injected animals (2% CR) than in control animals (Control, * *p* < 0.05). Animals treated with intrathecal capsazepine (CZP 1 μg) exhibited an attenuated increase in spinal glutamate concentration induced by carrageenan (# *p* < 0.05, Figure 4A).

Assessment of serum glutamate concentration revealed that glutamate levels were significantly increased after carrageenan injection (2% CR) and capsazepine administration (CZP 1 μg) compared to those in the control group (Control, * *p* < 0.05), but no significant differences were observed in serum glutamate concentration between capsazepine-treated mice and carrageenan-treated mice (Figure 4B).

### 2.5. Expression of TRPV1, pNR1, and pNR2B in the Spinal Dorsal Horn

Western blot data confirmed the effect of intrathecal capsazepine treatment on spinal TRPV1, pNR1, and pNR2B expression 1 h after carrageenan injection. The group of carrageenan injection induced spinal TRPV1 expression of 131.6 ± 20.31%, which was significantly higher than that in the control group (80.76 ± 6.38%, * *p* < 0.05). Intrathecal pretreatment with capsazepine significantly suppressed carrageenan-enhanced TRPV1 expression in the spinal dorsal horn (80.90 ± 6.62%, # *p* < 0.05, Figure 5A). The expression level of pNR1 in the spinal cord did not show a significant difference between the groups (Figure 5B). In the case of spinal pNR2B expression, the carrageenan injection group showed 131.0 ± 10.16%, which showed a significant increase compared to the control group (85.51 ± 5.83%, ** *p* < 0.01), and animals of intrathecal capsazepine treatment decreased carrageenan-enhanced spinal pNR2B expression (96.57 ± 10.57%, # *p* < 0.05, Figure 5C). Injection of carrageenan significantly increased the expression of TRPV1 and pNR2B, but not pNR1, in the spinal dorsal horn compared to that in control animals. Intrathecal pretreatment with capsazepine suppressed carrageenan-enhanced TRPV1 and pNR2B expression in the spinal dorsal horn, suggesting potent anti-nociceptive effects of capsazepine in carrageenan-induced inflammatory pain.

## 3. Discussion

Increased sensitivity of afferent neurons due to inflammatory response related to tissue damage is characterized by inducing hyperalgesia (the pain evoked by a noxious stimulus is exaggerated in both ampli-tude and duration) and allodynia (an innocuous stimulus is perceived as painful) [19]. Inflammation caused by noxious stimuli or disease is a normal biological reaction to injury that activates nociceptors and evokes protective behaviors to prevent tissue damage [5]. The present study demonstrated that intraplantar injection of carrageenan produced peripheral inflammatory pain responses (mechanical allodynia and thermal hyperalgesia) and intrathecal administration of capsazepine suppressed carrage18enan-induced peripheral inflammatory pain-related behaviors in mice. Using the capsaicin-induced spontaneous pain test, we determined the optimal dose at which intrathecal treatment with capsazepine produced anti-nociceptive effects. TRP channels play key roles in the regulation of inflammation via the release of neuropeptides and act through the modulation of intracellular pathways to trigger inflammatory mechanisms. TRP ankrin (TRPA), TRP melastatin (TRPM), and TRP vanilloid (TRPV) are TRP channels expressed on neurons that communicate with the immune system and peripheral organs to regulate inflammatory responses [3]; however, TRPV1 is the most well studied and its properties are well known.

TRPV1, a ligand-gated cation channel, is a key mediator in various types of pain, including inflammatory pain, and is widely expressed in pain-modulating areas of the peripheral and central nervous systems [20]. TRPV1 is activated by inflammatory endogenous mediators, and endothelin during inflammatory pain conditions [4]. Our behavioral experiments confirmed that intrathecal administration of capsazepine suppressed carrageenan-induced nociceptive behavior, suggesting that TRPV1 regulates carrageenan-induced inflammatory pain. In agreement with our findings, inhibition of TRPV1 has been identified as a potential target for the treatment of inflammatory pain. In a preclinical model of rheumatoid arthritis, depletion of TRPV1 positive cells reduced arthritis severity and depleted neuropeptide levels resulting from increased intracellular calcium [3]. Further, TRPV1 knockout mice exhibited reduced swelling of the knee joint and hyper-permeability [8]. Blocking peripheral TRPV1 using capsazepine before carrageenan injections reduced ipsilateral nociceptive behavior during the acute phase, and contralateral nociceptive behavior was almost completely abolished during both the acute and subacute phases [21]. Subcutaneous administration of capsazepine has been reported to reduce mechanical hyperalgesia in rodent models of inflammatory hyperalgesia [22]. Intraperitoneal injection of capsazepine attenuated heat hyperalgesia but did not affect mechanical and cold allodynia in a murine model of cancer pain [23]. In intrathecal administration experiments, capsazepine transiently reversed mechanical and heat allodynia increased by X-ray [24] and relieved remifentanil-induced postoperative hyperalgesia [25]. In another study, intrathecal administration of capsazepine decreased the analgesic effect against thermal hyperalgesia induced by anandamide, endogenous cannabinoid receptor ligand, in the carrageenan model [26].

Western blot analysis revealed that injection of carrageenan increased the expression of spinal TRPV1 and phosphorylation of NR2B, but not NR1. Intrathecal treatment with capsazepine inhibited carrageenan-enhanced TRPV1 expression in the spinal dorsal horn. This result is consistent with the results of our behavioral experiment that inhibition of spinal TRPV1 significantly reduced pain-related behaviors. TRPV1 has been shown to be associated with functions such as neurogenic inflammation, neuropathic pain, autoimmune disorders, cancer and immune cells, and many TRPV1 agonists (capsaicin or resiniferatoxin) and antagonists (capsazepine, BCTC, or SB-705498) have been tested to treat various related diseases [27]. Administration of capsazepine or TRPV1 siRNA attenuated airway inflammation, hypersensitiveness, as well as reduced levels of pro-inflammatory neuropeptides and oxidative stress in mouse model of chronic asthma [28,29]. Pre-treatment with capsazepine in lipopolysaccharide-activated macrophages significantly suppressed production of pro-inflammatory cytokines IL-6, IL-1b, and IL-18 and COX-2 expression [30]. In the glutamate assay test, administration of carrageenan increased the concentration of glutamate in the spinal cord and serum, whereas administration of capsazepine attenuated the carrageenan-induced increase in glutamate levels. Glutamate is a major excitatory neurotransmitter used by primary afferent synapses and neurons in the spinal cord. Glutamate and its receptors are widely expressed in areas of the central and peripheral nervous systems that participate in several neurophysiological functions, particularly pain sensation and transmission [31]. Peripherally released glutamate activates sensory neurons expressing glutamate receptors at their peripheral terminals, thereby triggering additional release of glutamate and other inflammatory mediators, resulting in exacerbation of inflammation [32]. NMDA receptors are ligand-gated cation channels activated by glutamate. These receptors are predominantly located at excitatory synapses and participate in excitatory neurotransmission in the central nervous system [33,34]. In physiological conditions, NMDA receptors are blocked by magnesium; as such, glutamate released in the spinal dorsal horn due to a noxious peripheral stimulus tends to interact with alpha-amino-3-hydroxy-5-methyl-4-isoxazolepropionic acid (AMPA) receptors. However, the magnesium block of NMDA receptors is removed in pathological conditions, and NMDA receptor activation is increased via several mechanisms, including NMDA receptor phosphorylation [11]. Exogenous NMDA applied to the dorsal spinal cord promoted the function of NMDA involved in nociception transmission, and D-isomer of AP5 exhibited significant inhibition of C-fiber-evoked responses of wide dynamic range (WDR) neurons for 20 h after induction of inflammation by carrageenan [35]. Administration of memantine, the NMDA receptor antagonist, showed significant analgesic effect in pre-treatment with carrageenan in animals with inflammatory pain caused by carrageenan, but no analgesic effect in post-treatment [36]. In particular, the results of the phosphorylation of the NMDA receptor in the process of pain transmission are still controversial. Our previous studies demonstrated that electrical stimulation around acupuncture points suppressed chemotherapy-induced neuropathy via modulation of spinal NR2B phosphorylation [15,16], and phosphorylation of spinal NR1 was regulated in animal models of neuropathic pain associated with chronic constrictive injury [37,38]. In the study of Robert et al., the lumbar spinal NR1 subunit was found to be phosphorylated within 2 h of the induction of inflammation, and the spinal NR2B expression was rather suppressed by carrageenan-induced inflammation [39]. Our results demonstrated that carrageenan-induced peripheral inflammatory pain was associated with an increase in phosphorylated NR2B levels in the spinal dorsal horn and glutamate levels in the spinal cord and serum. Thus, the phosphorylation of NMDA receptors is considered a key factor in the development and maintenance of pain caused by exogenous stimuli. Collectively, the findings suggest that glutamate and pNR2B levels in the spinal cord increased after carrageenan injection, and these increases were significantly suppressed by inhibition of TRPV1.

The functional relationship between TRPV1 and NMDA receptors in the regulation of inflammatory pain remains poorly understood. In a related study, NMDA receptors and TRPV1 were observed to form functional complexes via Ca^2+^ calmodulin-dependent protein kinase II (CaMKII) and protein kinase C (PKC) signaling cascades, contributing to the development of mechanical hyperalgesia in the masseter muscle in rats [40]. Another study reported that ginger extract and its compounds attenuated hyperalgesia and allodynia in mice by reducing spinal TRPV1 and NR2B expression in a streptozotocin-induced mouse model of painful diabetic neuropathy [6]. These results are consistent with the interactions of TRPV1 and NMDA receptors in the mouse spinal cord, which are essential for the development and transmission of inflammatory pain observed in the present study.

## 4. Materials and Methods

### 4.1. Animals

All experiments were performed using male ICR mice (20–25 g; Samtako, Osan, Korea) housed in colony cages with ad libitum access to food and water. Animals were maintained in a standard environment consisting of a 12 h light/dark cycle with lights on at 07:00, and constant room temperature (24 ± 2 °C) and humidity (40–60%) for at least 1 week prior to experimentation. All experiments were conducted in accordance with the ethical guidelines of the International Association for the Study of Pain and were approved by the Animal Care and Use Committee at the Korea Institute of Oriental Medicine (KIOM) (reference number: #20-034). Efforts were made to minimize animal distress and reduce the number of animals used in this study.

### 4.2. Carrageenan-Induced Inflammatory Pain

Peripheral inflammation was induced by intraplantar injection of 50 μL of 2% λ-carrageenan (Sigma, St. Louis, MO, USA) suspended in sterile saline solution into the right hind paw. The control animals received 50 μL sterile saline administered into the right hind paw. Intraplantar injection of carrageenan or saline was performed under light anesthesia with 3% isoflurane in a mixture of N_2_O/O_2_ gas. To examine carrageenan-induced nociceptive behaviors, animals (total n = 24) were randomly divided into three groups as follows: naïve animals (Control, n = 8), saline-injected animals (Vehicle, n = 8), and 2% carrageenan intraplantar-injected animal (2% CR, n = 8). To examine the analgesic effects of capsazepine, animals (total n = 24) were randomly divided into three treatment groups as follows: 2% carrageenan-injected animals (2% CR, n = 8), saline intrathecal-treated animals (Vehicle [i.t.], n = 8), and capsazepine intrathecal-treated animal (CZP [i.t.], n = 8). The intrathecal injection of capsazepine was performed 5 min before injection of carrageenan. This time point was selected because it is when the expression of carrageenan-induced inflammatory substances (e.g., IL-1β) in the spinal cord is the most highly upregulated as shown in our previous study [41].

### 4.3. Evaluation of Nociceptive Behaviors

All behavioral assessments were performed under the ethical guidelines set forth by the International Association for the Study of Pain (IASP). To assess carrageenan-induced peripheral inflammatory pain responses such as mechanical allodynia and thermal hyperalgesia, we measured paw withdrawal responses using a von Frey filament (0.4 g, North Coast Medical, CA, USA) and plantar analgesia meter (IITC Life Science Inc., Woodland Hills, CA, USA), as described previously [15]. Briefly, normal baseline values of the withdrawal response to mechanical or heat stimulation were measured prior to carrageenan injections. Mice were then randomly assigned to each treatment group, and behavioral testing was performed in a blinded manner. During the experimental period, all behavioral tests were performed at the following time points after carrageenan injection: 1, 2, 4, 8, and 24 h. Tests were conducted at the same time of day to reduce errors in relation to the diurnal rhythm.

Paw withdrawal response frequency (PWF) to normal innocuous mechanical stimuli was measured using a von Frey filament with a force of 0.4 g. Mice were placed on a metal mesh grid under a plastic chamber, and the von Frey filament was applied from underneath the metal mesh flooring to each plantar surface of the hind paw. The von Frey filament was applied 10 times to the hind paw, and the number of paw withdrawal responses out of 10 was quantified. The results of mechanical behavioral testing in each animal were expressed as withdrawal response frequency percentage (PWF, %), which represents the percentage of paw withdrawals out of a maximum of 10. To determine nociceptive responses to heat stimuli, animals were placed in a plastic chamber (15 cm in diameter and 20 cm in length) on a glass floor and allowed to acclimate for 30 min before thermal hyperalgesia testing. A radiant heat source was positioned under the glass floor beneath each hind paw, and paw withdrawal latency (WRL) was measured to the nearest 0.1 s using a plantar analgesia meter. The cut-off time was set at 20 s to prevent tissue damage. All behavioral tests were conducted by an experimenter who was blinded to the treatment conditions.

### 4.4. Capsaicin-Induced Spontaneous Pain Test and Drug Injection

To investigate the effect of the TRPV1 antagonist capsazepine on pain induced by TRPV1 channel activation, several doses of capsazepine were administered in a capsaicin-induced spontaneous pain model. The capsaicin (Sigma-Aldrich, St. Louis, MO, USA) test was performed as described previously [42]. Briefly, mice received intraplantar injections of 20 μL of capsaicin solution (1.6 μg/paw) into the right hind paw. Immediately after capsaicin injection, animals were placed into clear observation chambers (15 × 15 × 15 cm), and nociceptive responses were evaluated as the time spent licking the injected paw for 15 min. The vehicle group consisted of animals injected with intraplantar physiological saline solution.

Capsazepine (Sigma-Aldrich, St. Louis, MO, USA) was dissolved in physiological saline solution and intrathecally administered at three doses (0.1, 1, and 10 μg per mouse) 5 min prior to capsaicin injection. The control group consisted of animals that received intrathecal physiological saline solution. The experimenter was blinded to treatment conditions.

Intrathecal (i.t.) injections of capsazepine were performed as reported previously [16], using a 10 μL Hamilton syringe with a 30-gauge needle. Briefly, the mouse was held tightly between the thumb and middle finger at the level of both iliac crests, and the fifth lumbar spinous process was palpated with the index finger. The needle was inserted through the vertebral column into the lumbar 5 to lumbar 6 intervertebral space, and a tail flick response was considered indicative of a successful i.t. injection. Capsazepine was slowly injected over a 10 s period. The needle was carefully removed from the spinal cord. Following injection, the animals were immediately returned to the observation chamber.

### 4.5. Glutamate Concentration Measurements

To measure glutamate concentration, spinal cord and serum samples from control, carrageenan-injected, and capsazepine-treated animals were collected 1 h after intraplantar carrageenan injections, the timepoint at which the anti-nociceptive effect of capsazepine reached its peak. Glutamate concentrations were measured using a colorimetric glutamate assay kit (Abcam, Cambridge, UK), which identifies glutamate as a specific substrate and forms a color product at 450 nm. Spinal cord and serum samples were homogenized in RIPA buffer containing protease and phosphatase inhibitors. Homogenates were centrifuged for 20 min at 10,000× *g* at 4 °C to remove solid material, and the supernatants were transferred to new tubes. Supernatant aliquots were used for quantitative measurements of glutamate concentration. An automated ELISA reader (Bio-Rad Laboratories, Hercules, CA, USA) was used to read the color of the reaction.

### 4.6. Western Blot and Image Analysis

For Western blot analysis, all procedures were performed as described in our previous report [43]. Mice were anesthetized by injecting a combination of 2.5 mg of Zoletil 50 (Virbac Laboratories, Carros, France) and 0.47 mg of Rompun (Bayer Korea, Seoul, Korea) in saline. The spinal cord was extracted by pressure expulsion with air into an ice-cooled, saline-filled glass dish and snap-frozen in liquid nitrogen. To verify the location of the L4–L6 spinal cord segments for Western blot, we identified the attachment site of each spinal nerve in anesthetized mice. Spinal segments were separated into left and right halves under a neurosurgical microscope. The spinal cord was subsequently further subdivided into dorsal and ventral halves by cutting straight across the central canal laterally to a midpoint in the white matter. The dorsal horns of the right and left spinal cords were then used for Western blot analysis. L4–L5 segments of the spinal cord were homogenized with RIPA buffer (Cell Signaling, Beverly, MA, USA) containing protease inhibitor, phosphatase inhibitor, and 0.1% sodium dodecyl sulfate (SDS). Insoluble materials were removed by centrifugation at 12,000× *g* for 20 min at 4 °C. Protein concentrations were determined using the Bradford reagent (Bio-Rad Laboratories, Hercules, CA, USA). Spinal cord lysates were separated by 6% or 10% sodium dodecyl sulfate-polyacrylamide gel electrophoresis (SDS-PAGE) and then transferred to a nitrocellulose membrane. Nonspecific binding was blocked with 5% non-fat milk (BD Biosciences, Franklin Lakes, NJ, USA) in T-TBS and 8% bovine serum albumin (MP Biomedical, Auckland, New Zealand) for 30 min at room temperature. The membrane was then incubated with anti-TRPV1 antibody (1:500, Santa Cruz Biotechnology, Santa Cruz, CA, USA), anti-phospho-NMDA receptor NR1 antibody (1:1000, Millipore, Billerica, MA, USA), and anti-phospho-NMDA receptor NR2B antibody (1:1000, Millipore, Billerica, MA, USA) in blocking solution overnight at 4 °C. The membrane was washed three times with T-TBS (10 min per wash) and incubated with goat anti-mouse IgG horseradish peroxidase (1:2000; Calbiochem, Darmstadt, Germany) or goat anti-rabbit IgG horseradish peroxidase (1:2000; Calbiochem, Darmstadt, Germany) for 1 h at room temperature. After the membrane was washed three times in T-TBS, the antibody reaction was visualized using a chemiluminescence assay kit (Pharmacia-Amersham, Freiburg, Germany). The intensity of the protein bands was analyzed using Image J software (Graph Pad Software, Stapleton, NY, USA, 2010).

### 4.7. Data Analysis

All data are expressed as the mean ± standard error of the mean (SEM) and were statistically analyzed using Prism 5.0 (GraphPad Software, San Diego, CA, USA). Data from behavioral studies were analyzed using two-way analysis of variance (ANOVA) to determine overall effects, and post-hoc analysis was performed using Tukey’s multiple comparisons test to determine *p*-values for experimental groups. Capsaicin and glutamate level data were analyzed using one-way ANOVA, followed by Tukey’s post-hoc analysis. For Western blot analysis, column analysis was performed using Student’s *t*-test for comparisons between two mean values. Differences were considered statistically significant at *p* < 0.05.

## 5. Conclusions

In conclusion, the present study demonstrated that intrathecal administration of a TRPV1 antagonist significantly attenuated carrageenan-induced mechanical allodynia and thermal hyperalgesia and suppressed the concentration of glutamate and pNR2B in the spinal cord of mice with inflammatory pain. Collectively, we presume that TRPV1 and NMDA receptors in the spinal cord are implicated in the transmission of inflammatory pain, and inhibition of TRPV1 may attenuate inflammatory pain via NMDA receptors. Indeed, the role of TRPV1 in various inflammatory disorders is increasingly being recognized. The current findings support the potential of using TRPV1 as a therapeutic target for inflammatory pain, although further mechanistic studies are warranted.

## Figures and Tables

**Figure 1 ijms-22-11177-f001:**
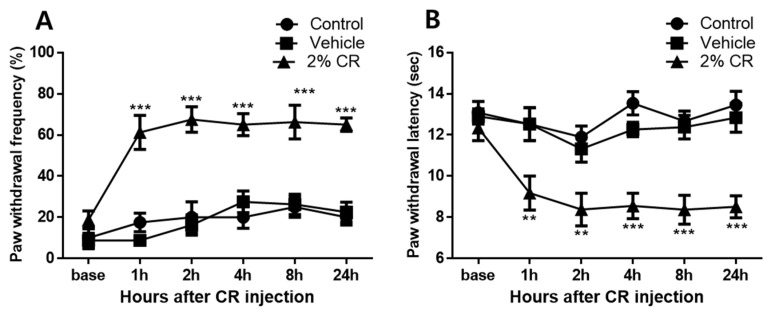
Graphs show the development of mechanical allodynia and thermal hyperalgesia in ipsilateral hind paws with 2% carrageenan (intraplantar) treatment. (**A**) No significant changes in paw withdrawal frequency (%) were observed in saline-treated animals (Vehicle, n = 8) over the 24 h experimental period compared to that in control animals (Control, n = 8). In contrast, the carrageenan injection group (2% CR, n = 8) exhibited a significant increase in paw withdrawal frequency in the hind limbs from 1 to 24 h after carrageenan injection, which underscores the robust development of mechanical allodynia induced by carrageenan (*** *p* < 0.001 compared to control mice). (**B**) No significant changes in paw withdrawal latency (sec) were observed in saline-treated animals over the 24 h experimental period compared to that in control animals. Carrageenan injection significantly decreased paw withdrawal latency from 1 to 24 h in the ipsilateral hind paw (** *p* < 0.01, *** *p* < 0.001 compared to control mice).

**Figure 2 ijms-22-11177-f002:**
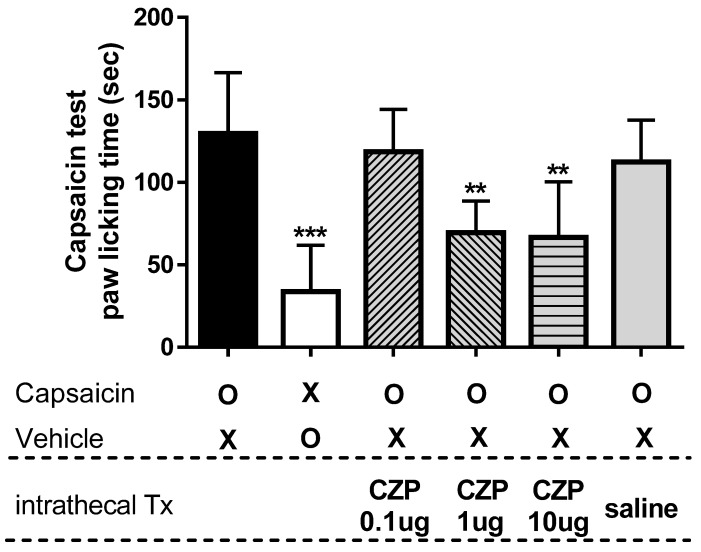
The graph shows the paw-licking time after intrathecal capsazepine administration in a mouse model of capsaicin-induced spontaneous pain. Intraplantar injection of capsaicin (capsaicin, n = 6) induced paw-licking behavior characteristic of nociception, while saline-injected animals (Vehicle, n = 6) exhibited significantly less paw-licking behavior compared to capsaicin-injected animals (*** *p* < 0.001). Doses of 1 and 10 μg of capsazepine (CZP 1 μg and CZP 10 μg, n = 6, respectively) significantly suppressed capsaicin-induced paw-licking time compared to that in capsaicin-injected mice (** *p* < 0.01), whereas a dose of 0.1 μg of capsazepine (CZP 0.1 μg, n = 6) and intrathecal administration of sterile saline (saline, n = 6) did not result in significant changes in paw-licking time.

**Figure 3 ijms-22-11177-f003:**
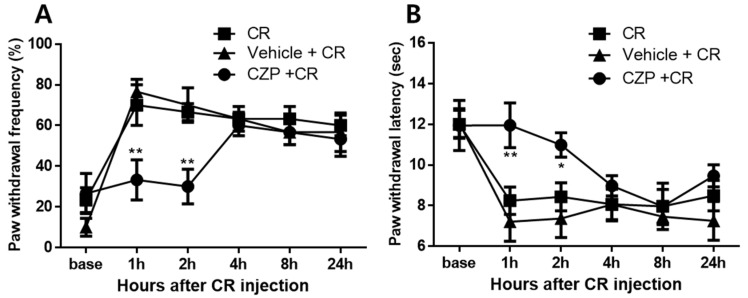
Graphs show the anti-nociceptive effects of intrathecal administration of 1 μg of capsazepine on mechanical allodynia (**A**) and thermal hyperalgesia (**B**) in mice with carrageenan-induced inflammatory pain. Carrageenan-induced mechanical allodynia at 1 and 2 h was significantly suppressed in the capsazepine-administered group (CZP + CR, n = 8) compared to that in mice with carrageenan-induced inflammatory pain (CR, n = 8). For thermal hyperalgesia, paw withdrawal latency increased from 1 to 2 h after carrageenan injection compared to that in carrageenan-injected mice (* *p* < 0.05, ** *p* < 0.01). No significant differences in paw withdrawal frequency (%) and paw withdrawal latency (sec) were observed between the vehicle group (Vehicle + CR, n = 8) and 2% carrageenan-injected animals at all timepoints of behavioral measurement.

**Figure 4 ijms-22-11177-f004:**
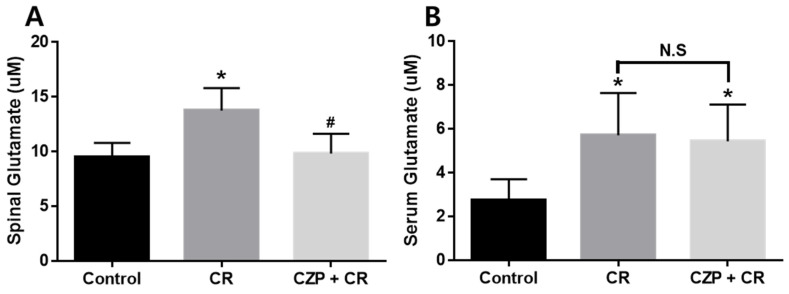
Graphs show the effect of intrathecal capsazepine treatment on the suppression of glutamate concentration in the spinal cord (**A**) and serum (**B**) of mice with carrageenan-induced inflammatory pain. Spinal glutamate concentration was significantly higher in carrageenan-injected animals (CR, n = 5) than in the control group (Control, n = 5, *****
*p* < 0.05). Glutamate concentration in the spinal cord was significantly lower in intrathecal capsazepine-treated animals (CZP + CR, n = 5) than in the carrageenan-injected group (# *p* < 0.05). Serum glutamate concentration was significantly higher in carrageenan-injected mice (CR, n = 5) and capsazepine-administered mice (CZP + CR, n = 5) than in control mice (Control, *****
*p* < 0.05). No significant differences (N.S) were observed in glutamate concentration between intrathecal capsazepine-treated mice and the carrageenan-treated group.

**Figure 5 ijms-22-11177-f005:**
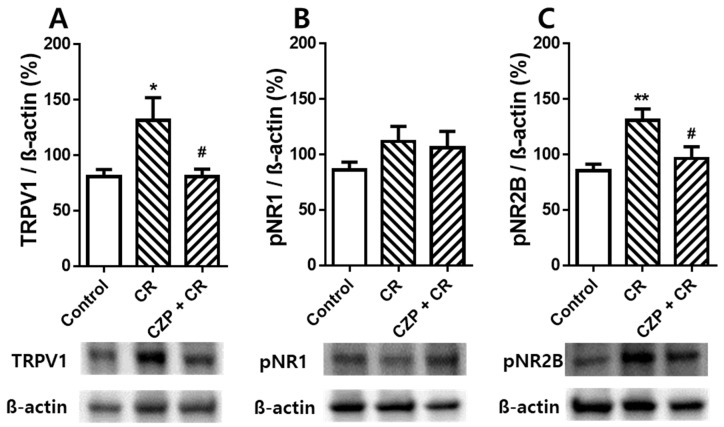
Graphs show the effects of intrathecal capsazepine treatment on TRPV1 (**A**), pNR1 (**B**), and pNR2B (**C**) expression in the lumbar spinal cord of mice with carrageenan-induced inflammation. TRPV1 expression in the spinal cord was significantly higher in carrageenan-injected animals (CR, n = 6) than in the control group (Control, n = 6, *****
*p* < 0.05) and was significantly lower in intrathecal capsazepine-treated animals (CZP + CR, n = 6) than in the carrageenan-injected group (# *p* < 0.05). pNR1 expression levels in the spinal cord were not significantly different among control, carrageenan-injected, and capsazepine-treated groups. Spinal pNR2B expression increased following carrageenan injection compared to that in control rats (******
*p* < 0.01), and intrathecal capsazepine-treated mice exhibited significantly lower spinal pNR2B expression compared to that in the carrageenan-treated group (# *p* < 0.05).

## Data Availability

Not applicable.

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
