# Peer review of "Inhibition of Spinal TRPV1 Reduces NMDA Receptor 2B Phosphorylation and Produces Anti-Nociceptive Effects in Mice with Inflammatory Pain"

_ijms, 2021, doi:10.3390/ijms222011177_

Round 1

Reviewer 1 Report

The manuscript: “Inhibition of spinal TRPV1 reduces NMDA receptor 2B phosphorylation and produces anti-nociceptive effects in mice with inflammatory pain” by Kang et al describe the ability of capsazepine to reduce the inflammatory pain induced by carrageenan and the interaction between TRPV1 and NMDA receptors.

Some major issues need to be addressed

The sentence at line 56-58 is poorly clear, what amino acids are at C-terminal of NMDA receptors? Those present in calmodulin-dependent kinase or PKA or PKC?

Moreover, the consistency with the cited references needs to be verified. For example, in ref 15 only the association between TRPV1 and calmodulin-dependent kinase is described and not the presence of amino acid residues of calmodulin in NMDA C terminal region.

The ref 7, in Discussion section, lane 202, is not consistent with the sentence reported: “In a preclinical model…..”

A check throughout the text is required to ascertain the correspondence between sentences and references.

In Fig 5A the comparison between Normal and CR has a statistical significance corresponding to p<0.05, in Fig 5C the comparison between Normal and CR has p<0.01, but the difference in percentage is exactly the same. This should be explained.

Author Response

Review Report (Reviewer 1)

The manuscript: “Inhibition of spinal TRPV1 reduces NMDA receptor 2B phosphorylation and produces anti-nociceptive effects in mice with inflammatory pain” by Kang et al describe the ability of capsazepine to reduce the inflammatory pain induced by carrageenan and the interaction between TRPV1 and NMDA receptors.

Some major issues need to be addressed

The sentence at line 56-58 is poorly clear, what amino acids are at C-terminal of NMDA receptors? Those present in calmodulin-dependent kinase or PKA or PKC? Moreover, the consistency with the cited references needs to be verified. For example, in ref 15 only the association between TRPV1 and calmodulin-dependent kinase is described and not the presence of amino acid residues of calmodulin in NMDA C terminal region.

Response to comment 1 of the Reviewer #1:  

è We made a mistake in expressions and references in the writing of the Introduction section. According to the reviewer’s comment, we rechecked the references and further revised the description of the NMDA receptors in the Introduction part as following; “The NMDA receptors composed of 4 subunits derived from the related families of NR1, NR2, and NR3, the typical NMDA receptors consist of two NR1 subunits that bind glycine and two NR2 subunits that bind glutamate [Curr Opin Pharmaco. 2007 Feb;7(1):39-47]. The NR1 is an essential subunit that combines with NR2 or NR3 subunits to form a functional receptor [Neurology. 2011 May 17;76(20):1750-7], and has been implicated in inflammatory pain sensitization during inflammation-induced nociception [Anesthesiology. 2010 Jun;112(6):1482-93]”, “The intracellular domains of NMDA receptor subunits contain consensus phosphorylation sites that regulates NMDA receptors for serine and threonine kinases. The two most actively studied NMDA receptors modulation are protein kinase A (PKA) and protein kinase C (PKC). The calcium/calmodulin-dependent protein kinase II (CAMKII) is also known to translocate to NMDARs in an activity-dependent manner.” [In: Biology of the NMDA Receptor. Van Dongen, A. M., Ed. Boca Raton (FL), 2009] (see page 2, line 52-64).

The ref 7, in Discussion section, lane 202, is not consistent with the sentence reported: “In a preclinical model…..” A check throughout the text is required to ascertain the correspondence between sentences and references.

Response to comment 2 of the Reviewer #1:  

è We made a mistake in marking references in the Discussion section. According to the reviewer’s comment, we rechecked the reported sentences and references, and marked matching references. [Front Immunol. 2020 Oct 23;11:590261.] (see page 7, line 214-216).

In Fig 5A the comparison between Normal and CR has a statistical significance corresponding to p<0.05, in Fig 5C the comparison between Normal and CR has p<0.01, but the difference in percentage is exactly the same. This should be explained.

Response to comment 3 of the Reviewer #1:  

è According to the reviewer's opinion, we analyzed again using the statistical program; Prism 5.0 (GraphPad Software, CA, USA). The figure showing the expression level of TRPV1 in spinal cord (Fig. 5A) shows the following results; The Mean ± SEM of the Normal group was 80.76 ± 6.38, and the Mean ± SEM of the CR group was 131.6 ± 20.31. The difference between the two Mean was 50.81 ± 21.29, and the P value was 0.0343(*p<0.05). The figure showing the expression level of pNR2B in spinal cord (Fig. 5C) shows the following results; The Mean ± SEM of the Normal group was 85.51 ± 5.83, and the Mean ± SEM of the CR group was 131.0 ± 10.16. The difference between the two Mean was 45.49 ± 11.71, and the P value was 0.0022(**p<0.01).

In order to show the results more clearly, we have added the supplementary explanation of western blot results for intrathecal injection of capsazepine in the Results section as following; “The group of carrageenan injection induced spinal TRPV1 expression of 131.6 ± 20.31 %, which was significantly higher than that in the normal group (80.76 ± 6.38 %, *p < 0.05). Intrathecal pretreatment with capsazepine significantly suppressed carrageenan-enhanced TRPV1 expression in the spinal dorsal horn (80.90 ± 6.62 %, #p < 0.05, Figure 5A). The expression level of pNR1 in the spinal cord did not show a significant difference between the groups (Figure 5B). In the case of spinal pNR2B expression, the carrageenan injection group showed 131.0 ± 10.16 %, which showed a significant increase compared to the normal group (85.51 ± 5.83 %, **p < 0.01), and animals of intrathecal capsazepine treatment decreased carrageenan-enhanced spinal pNR2B expression (96.57 ± 10.57 %, #p < 0.05, Figure 5C)” (see page 6, line 165-174).

Reviewer 2 Report

I read the study of Kang et al. with great interest. In this study, the authors showed an increase of spinal TRPV1 receptors in carrageenan-induced inflammatory pain model animals. According to the experiments of the authors, the carrageenan treatment also increased the spinal glutamate levels and phosphorylation of NMDA receptor subunit NR2B but not NR1. The authors also demonstrated that intrathecal administration of capsazepine, a TRPV1 antagonist, could prevent the increase of glutamate levels and NR2B phosphorylation and thus inhibited pain behaviors in the carrageenan model. Overall, the study was appropriately designed and performed with rigorous methods. The findings of the authors are significant and interesting, and the manuscript is well-written. I have only a few remarks.

  1. The authors examined only two subunits of NMDA receptor (NR1 and NR2B) phosphorylation. What about total levels? Was there any reason to assess the phosphorylation of these two types but not others such as NR2A? I recommend the authors add some rationale and references in their manuscript regarding these.
  2. In figure 5A, the authors showed that the TRPV1 receptor expression was 1.5 fold higher in carrageenan-treated mice than control level. More interestingly, the increased expression of TRPV1 was reversed by the treatment of TRPV1 blocker capsazepine. The alteration and reverse of the spinal TRPV1 expression in these conditions deserve attention and possible underlying mechanisms should be further discussed in the manuscript.
  3. In figure 5B, the representative samples of western blot are not matching well with the bar graph. To me, it seems that the normal pNR1 level is very low compared to the CR-treated two conditions when it comes to gel loading.

Author Response

Review Report (Reviewer 2)

I read the study of Kang et al. with great interest. In this study, the authors showed an increase of spinal TRPV1 receptors in carrageenan-induced inflammatory pain model animals. According to the experiments of the authors, the carrageenan treatment also increased the spinal glutamate levels and phosphorylation of NMDA receptor subunit NR2B but not NR1. The authors also demonstrated that intrathecal administration of capsazepine, a TRPV1 antagonist, could prevent the increase of glutamate levels and NR2B phosphorylation and thus inhibited pain behaviors in the carrageenan model. Overall, the study was appropriately designed and performed with rigorous methods. The findings of the authors are significant and interesting, and the manuscript is well-written. I have only a few remarks.

  1. The authors examined only two subunits of NMDA receptor (NR1 and NR2B) phosphorylation. What about total levels? Was there any reason to assess the phosphorylation of these two types but not others such as NR2A? I recommend the authors add some rationale and references in their manuscript regarding these.

Response to comment 1 of the Reviewer #2: 

è NMDA receptors are heteromeric complexes incorporating different subunits within a repertoire of three subtypes: NR1, NR2 and NR3. There are eight different NR1 subunits generated by alternative splicing from a single gene, four different NR2 subunits (A, B, C and D) and two NR3 subunits (A and B); the NR2 and NR3 subunits [Curr Opin Pharmaco. 2007 Feb;7(1):39-47]. The typical NMDA receptor requires 2 NR1 subunits, which bind glycine, and 2 NR2 subunits, which bind glutamate. The NR1 is an essential subunit that combines with NR2 or NR3 subunits to form a functional receptor, and has been implicated in inflammatory pain sensitization during inflammation-induced nociception [Neurology. 2011 May 17;76(20):1750-7]. In the case of NR2B subunit, there are many research results that it is a receptor subunit related to pain. Recently, we demonstrated that electrical stimulation suppressed the expression of phosphorylated NR2B in the spinal cord in chemotherapy-induced peripheral neuropathic pain [The American journal of Chinese medicine 2015, 43, (1), 57-70, Brain research bulletin 2020, 162, 237-244].

According to the reviewer’s comment, we rechecked the references and further revised the description of the NMDA receptors in the introduction part as following; “The NMDA receptors composed of 4 subunits derived from the related families of NR1, NR2, and NR3, the typical NMDA receptors consist of two NR1 subunits that bind glycine and two NR2 subunits that bind glutamate [Curr Opin Pharmaco. 2007 Feb;7(1):39-47]. The NR1 is an essential subunit that combines with NR2 or NR3 subunits to form a functional receptor [Neurology. 2011 May 17;76(20):1750-7], and has been implicated in inflammatory pain sensitization during inflammation-induced nociception [Anesthesiology. 2010 Jun;112(6):1482-93]”, “The intracellular domains of NMDA receptor subunits contain consensus phosphorylation sites that regulates NMDA receptors for serine and threonine kinases. The two most actively studied NMDA receptors modulation are protein kinase A (PKA) and protein kinase C (PKC). The calcium/calmodulin-dependent protein kinase II (CAMKII) is also known to translocate to NMDARs in an activity-dependent manner.” [In: Biology of the NMDA Receptor. Van Dongen, A. M., Ed. Boca Raton (FL), 2009] (see page 2, line 52-64).

  1. In figure 5A, the authors showed that the TRPV1 receptor expression was 1.5 fold higher in carrageenan-treated mice than control level. More interestingly, the increased expression of TRPV1 was reversed by the treatment of TRPV1 blocker capsazepine. The alteration and reverse of the spinal TRPV1 expression in these conditions deserve attention and possible underlying mechanisms should be further discussed in the manuscript.

Response to comment 2 of the Reviewer #2: 

è TRPV1 was the first identified member of the vanilloid receptor subfamily of TRP ion channels and to date is the most extensively studied. It was characterized in 1997 by Julius group as a receptor of capsaicin, a pungent compound from chili peppers [Nature, 1997 Oct 23;389(6653):816-24]. TRPV1, known as a capsaicin receptor, is widely expressed in peripheral and central nervous system regions involved in pain transmission and modulation. It is activated by inflammatory mediators such as protease, ATP, and bradykinin under inflammatory pain conditions and is upregulated in neuropathic pain conditions. TRPV1 inhibition by capsazepine regulated inflammation by inhibiting various inflammation-related substances in various diseases.

Following the comments of the reviewers, we added some possible underlying mechanisms to the Discussion section, citing references: “Western blot analysis revealed that injection of carrageenan increased the expression of spinal TRPV1 and phosphorylation of NR2B, but not NR1. Intrathecal treatment with capsazepine inhibited carrageenan-enhanced TRPV1 expression in the spinal dorsal horn. These result is consistent with the results of our behavioral experiment that inhibition of spinal TRPV1 significantly reduced pain-related behaviors. TRPV1 has been shown to be associated with functions such as neurogenic inflammation, neuropathic pain, autoimmune disorders, cancer and immune cells, and many TRPV1 agonists (capsaicin or resiniferatoxin) and antagonists (capsazepine, BCTC, or SB-705498) have been tested to treat various related diseases [Front Oncol, 2019 Oct 16;9:1087]. Administration of capsazepine or TRPV1 siRNA attenuated airway inflammation, hypersensitiveness, as well as reduced levels of pro-inflammatory neuropeptides and oxidative stress in mouse model of chronic asthma [Sci Rep, 2017 Sep 20;7(1):11926, Allergy Asthma Immunol Res, 2018 May;10(3):216-224]. Pre-treatment with capsazepine in lipopolysaccharide-activated macrophages significantly suppressed production of pro-inflammatory cytokines IL-6, IL-1b, and IL-18 and COX-2 expression [Biochem Biophys Res Commun, 2017 Mar 11;484(3):668-674].” (see page 7, line 226-243).

Despite that in many studies capsazepine was used as a TRPV1 antagonist, the result has to be treated with caution. Capsazepine recently was found to activate TRPA1 channel, which along with TRPV1 is involved in nociception and inflammation. Kistner et al. [Sci Rep, 2016 Jun 30;6:28621] found that capsazepine administration led to attenuation of inflammation in the model of murine colitis. Nevertheless, authors demonstrated that the effect was TRPV1-independent and rather associated with desensitization of TRPA1. Based on the findings of the Kistner's group, we are currently planning additional experiments on TRPV1 as well as its association with TRPA channel.

  1. In figure 5B, the representative samples of western blot are not matching well with the bar graph. To me, it seems that the normal pNR1 level is very low compared to the CR-treated two conditions when it comes to gel loading.

Response to comment 3 of the Reviewer #2: 

è Based on the reviewer's comments, we replaced the representative sample picture of pNR1 in Figure 5B in the Results section. (see page 6, Figure 5).

In addition, in the case of TRPV1 and pNR2B, which show statistically significant differences, we added supplementary explanations to clearly show the results in the Results section as following; “The group of carrageenan injection induced spinal TRPV1 expression of 131.6 ± 20.31 %, which was significantly higher than that in the normal group (80.76 ± 6.38 %, *p < 0.05). Intrathecal pretreatment with capsazepine significantly suppressed carrageenan-enhanced TRPV1 expression in the spinal dorsal horn (80.90 ± 6.62 %, #p < 0.05, Figure 5A). The expression level of pNR1 in the spinal cord did not show a significant difference between the groups (Figure 5B). In the case of spinal pNR2B expression, the carrageenan injection group showed 131.0 ± 10.16 %, which showed a significant increase compared to the normal group (85.51 ± 5.83 %, **p < 0.01), and animals of intrathecal capsazepine treatment decreased carrageenan-enhanced spinal pNR2B expression (96.57 ± 10.57 %, #p < 0.05, Figure 5C)” (see page 6, line 164-173).

Reviewer 3 Report

Here Kang et al. seek to evaluate the impact of TRPV1 inhibition via intrathecal administration of the antagonist capsazepine on inflammatory pain and NMDA receptor phosphorylation in the carrageenan-induced paw edema model. While I find no fault with the dataset, my biggest concern in the paucity of new insight into these mechanism(s). The majority of data presented are simple repeats of prior work, and the new data linking TRPV1 to NMDAR phosphorylation is observational and lacks evidence of functional significance. Similarly, citations for several key studies relevant to this work are missing from the discussion. In my mind, additional data must be provided before I would deem this work suitable for publication.

Major Concerns:

  • The ability of intrathecal capsazepine to decrease thermal and mechanical hyperalgesia in the carrageenan model has been previously established in rats(1) and confirmed again in TRPV1 knockout mice(2). While the datasets presented in figures 1-3 are important to establish the veracity of the model, there is no new information presented in these datasets. Thus, the new data presented in this paper are found in figures 4-5. Here the authors establish that inhibition of TRPV1 channels in spinal cord decreases spinal glutamate and NR2B phosphorylation. While these biochemical changes do appear to depend on TRPV1, what remains unknown is if they matter for the therapeutic impacts of capsazepine. Barring data explicitly demonstrating the importance of NMDARs in inflammatory pain, this paper has little in the way of novel insight into the anti-nociceptive actions of TRPV1 inhibition. Prior work has shown an increase in spinal NMDAR activity following carrageenan administration(3), and the NMDAR antagonist memantine inhibits carrageenan-dependent hyperalgesia when administered prophylactically but not therapeutically(4). This issue must be addressed. I have a few questions/suggestions:
    1. The timing of NMDA inhibition in this model is critical. Memantine only provides anti-nociception if given before carrageenan implantation. In the methods section, the authors do not state the timing of capsazepine administration. Was it before, during, or after carrageenan implantation? If after, the impacts of NMDAR are probably minimal.
    2. Presumably, if TRPV1 antagonism is acting via NMDAR inhibition, the addition of a drug like memantine, while efficacious on its own, would provide no additional protection if co-administered with capsazepine. The authors should test this.
    3. The authors should add the papers I’ve cited (1-4) to their discussion.
  • The impacts on NR1 and NR2 subunits seen in this work following carrageenan injection are in direct contradiction to a prior study in rats(5). E.g. those authors observed an increase in NR1 phosphorylation and a decrease in NR2B levels with no change in NR2B phosphorylation. Here, Kang et al. observe the opposite trends. The authors should discuss and speculate as to why their data differ so greatly from prior work.

Minor concerns:

  • For figures 3-5, the authors should indicate that the “vehicle” and “CZP” groups also receive a CR injection. This is implied, but should be explicitly stated in each graph to prevent confusion
    1. Figure 3 groups: CR, Vehicle + CR, CZP + CR
    2. Figure 4 groups: Control, CR, CZP + CR
    3. Figure 5 groups: Control, CR, CZP + CR
  • Abstract line 11: “is an important mediator of various noxious stimuli” is generic. TRPV1 is a mediator of the inflammatory response to various noxious stimuli.
  • Introduction line 56: The sentence “NMDA receptors comprise several amino acid residues at the C-terminal, including calcium/calmodulin-dependent protein kinase II (CaMKII), protein kinase A (PKA), and protein kinase C (PKC)” does not make sense. I think the authors meant to say that NMDA receptors are “regulated by kinases at several amino resides on the C-terminus”
  • Referring to a group as “normal” seems strange. Better to call them the “control” group.
  • Conclusion line 386: “TRPV1 attenuates inflammatory pain via NMDA receptors” is an overstatement. TRPV1 MAY do this, but the authors provide no evidence.

References

  • Horvath G, Kekesi G, Nagy E, and Benedek G. The role of TRPV1 receptors in the antinociceptive effect of anandamide at spinal level. Pain. 2008;134(3):277-84.
  • Watanabe M, Ueda T, Shibata Y, Kumamoto N, and Ugawa S. The role of TRPV1 channels in carrageenan-induced mechanical hyperalgesia in mice. Neuroreport. 2015;26(3):173-8.
  • Rygh LJ, Svendsen F, Hole K, and Tjolsen A. Increased spinal N-methyl-D-aspartate receptor function after 20 h of carrageenan-induced inflammation. Pain. 2001;93(1):15-21.
  • Eisenberg E, LaCross S, and Strassman AM. The effects of the clinically tested NMDA receptor antagonist memantine on carrageenan-induced thermal hyperalgesia in rats. Eur J Pharmacol. 1994;255(1-3):123-9.
  • Caudle RM, Perez FM, Del Valle-Pinero AY, and Iadarola MJ. Spinal cord NR1 serine phosphorylation and NR2B subunit suppression following peripheral inflammation. Mol Pain. 2005;1:25.

Author Response

Review Report (Reviewer 3)

Here Kang et al. seek to evaluate the impact of TRPV1 inhibition via intrathecal administration of the antagonist capsazepine on inflammatory pain and NMDA receptor phosphorylation in the carrageenan-induced paw edema model. While I find no fault with the dataset, my biggest concern in the paucity of new insight into these mechanism(s). The majority of data presented are simple repeats of prior work, and the new data linking TRPV1 to NMDAR phosphorylation is observational and lacks evidence of functional significance. Similarly, citations for several key studies relevant to this work are missing from the discussion. In my mind, additional data must be provided before I would deem this work suitable for publication.

Major Concerns:

  • The ability of intrathecal capsazepine to decrease thermal and mechanical hyperalgesia in the carrageenan model has been previously established in rats(1) and confirmed again in TRPV1 knockout mice(2). While the datasets presented in figures 1-3 are important to establish the veracity of the model, there is no new information presented in these datasets. Thus, the new data presented in this paper are found in figures 4-5. Here the authors establish that inhibition of TRPV1 channels in spinal cord decreases spinal glutamate and NR2B phosphorylation. While these biochemical changes do appear to depend on TRPV1, what remains unknown is if they matter for the therapeutic impacts of capsazepine. Barring data explicitly demonstrating the importance of NMDARs in inflammatory pain, this paper has little in the way of novel insight into the anti-nociceptive actions of TRPV1 inhibition. Prior work has shown an increase in spinal NMDAR activity following carrageenan administration(3), and the NMDAR antagonist memantine inhibits carrageenan-dependent hyperalgesia when administered prophylactically but not therapeutically(4). This issue must be addressed. I have a few questions/suggestions:

  1. The timing of NMDA inhibition in this model is critical. Memantine only provides anti-nociception if given before carrageenan implantation. In the methods section, the authors do not state the timing of capsazepine administration. Was it before, during, or after carrageenan implantation? If after, the impacts of NMDAR are probably minimal.

Response to Major comment 1-1 of the Reviewer #3: 

è We made the mistake of omitting the timing of intrathecal administration of capsazepine in the Methods section. We administered capsazepine intrathecally 5 minutes prior to carrageenan injection. We have used the intrathecal administration method in many previous studies, and in most studies, a single intrathecal drug administration was performed for 5 minutes to see the effect of the administered drug [Brain Res Bull, 2011 Nov 25;86(5-6):412-21], [J Pain, 2012 Feb;13(2):155-66], [Pain, 2015 Jun;156(6):1046-1059], [Exp Neurol, 2017 Jan;287(Pt 1):1-13].

Based on the reviewer's questions/suggestions, we added intrathecal administration of capsazepine time in the Method section as following; “The intrathecal injection of capsazepine was performed 5 minutes before injection of carrageenan. This time point was selected because it is when the expression of carrageenan-induced inflammatory substances (e.g. IL-1β) in the spinal cord is the most highly upregulated as shown in our previous study [Pain, 2015 Jun;156(6):1046-1059].” (see page 9, line 317-320).

  1. Presumably, if TRPV1 antagonism is acting via NMDAR inhibition, the addition of a drug like memantine, while efficacious on its own, would provide no additional protection if co-administered with capsazepine. The authors should test this.

Response to Major comment 1-2 of the Reviewer #3: 

è This study was an experiment to investigate the relevance of whether TRPV1 inhibition using capsazepine activates the NMDA receptor. As a result, capsazepine treatment inhibited phosphorylation of NR2B.

In fact, we experimented a capsaicin test using the MK-801, NMDA receptor antagonist. As a result, MK-801 strongly inhibited capsaicin-induced pain. However, experiments on the co-administration of TRPV1 inhibitor and NMDA inhibitor have not yet been conducted. The reason is that the result of co-administration is a result necessary for the next paper we are additionally in progress, and it was judged that it does not fit the flow of this study. We are currently conducting an experiment on the analgesic efficacy and its mechanism of acupuncture stimulation as an alternative medicine, focusing on the relevance of TRPV1, TRPA and NMDA, and would like to submit the next paper. The graph of the experimental results for MK-801 and acupuncture stimulation in the capsaicin test are as follows.

  1. The authors should add the papers I’ve cited (1-4) to their discussion.

Response to Major comment 1-3 of the Reviewer #3: 

è Based on the reviewers' comment, we added references requested by the reviewers to the Discussion section. Among the two references related to TRPV1, one has already been cited [Neuroreport, 2015 Feb 11;26(3):173-8], and the other has been added as follows; “Blocking peripheral TRPV1 using capsazepine before carrageenan injections reduced ipsilateral nociceptive behavior during the acute phase, and contralateral nociceptive behavior was almost completely abolished during both the acute and subacute phases [Neuroreport, 2015 Feb 11;26(3):173-8].” (see page 7, line 217-220). “In another study, intrathecal administration of capsazepine decreased the analgesic effect against thermal hyperalgesia induced by anandamide, endogenous cannabinoid receptor ligand, in the carrageenan model [Pain, 2008 Feb;134(3):277-284].” (see page 7, line 226-229).

Two references requested by reviewers related to NMDA receptor were also added as follows; “Exogenous NMDA applied to the dorsal spinal cord promoted the function of NMDA involved in nociception transmission, and D-isomer of AP5 exhibited significantly inhibition of C-fiber-evoked responses of wide dynamic range (WDR) neurons for 20 hours after induction of inflammation by carrageenan [Pain, 2001 Jul;93(1):15-21]. Administration of memantine, the NMDA receptor antagonist, showed significant analgesic effect in pre-treatment with carrageenan in animals with inflammatory pain caused by carrageenan, but no analgesic effect in post-treatment [Eur J Pharmacol, 1994 Apr 1;255(1-3):123-9]. (see page 8, line 261-267).

  • The impacts on NR1 and NR2 subunits seen in this work following carrageenan injection are in direct contradiction to a prior study in rats(5). g. those authors observed an increase in NR1 phosphorylation and a decrease in NR2B levels with no change in NR2B phosphorylation. Here, Kang et al. observe the opposite trends. The authors should discuss and speculate as to why their data differ so greatly from prior work.

Response to Major comment 2 of the Reviewer #3: 

è The NMDA receptors composed of 4 subunits derived from the related families of NR1, NR2, and NR3, the typical NMDA receptors consist of two NR1 subunits that bind glycine and two NR2 subunits that bind glutamate [Curr Opin Pharmacol, 2007 Feb;7(1):39-47]. Among these subunits, there are many references that NR1 and NR2B are involved in pain transmission. Therefore, we assumed that the phosphorylation of NR1 and NR2B would affect the transmission of inflammatory pain by carrageenan and proceeded with the experiment. Our results showed that NR1 had no effect and that NR2B was involved in inflammatory pain transmission. However, opinions about the involvement of the NMDA receptor subunits are still controversial.

Based on the reviewer's comments, we added the following to the Introduction and Discussion section about the activation of the NMDA receptor subunits; “The NMDA receptors composed of 4 subunits derived from the related families of NR1, NR2, and NR3, the typical NMDA receptors consist of two NR1 subunits that bind glycine and two NR2 subunits that bind glutamate [Curr Opin Pharmaco. 2007 Feb;7(1):39-47]. The NR1 is an essential subunit that combines with NR2 or NR3 subunits to form a functional receptor [Neurology. 2011 May 17;76(20):1750-7], and has been implicated in inflammatory pain sensitization during inflammation-induced nociception [Anesthesiology. 2010 Jun;112(6):1482-93]”, “The intracellular domains of NMDA receptor subunits contain consensus phosphorylation sites that regulates NMDA receptors for serine and threonine kinases. The two most actively studied NMDA receptors modulation are protein kinase A (PKA) and protein kinase C (PKC). The calcium/calmodulin-dependent protein kinase II (CAMKII) is also known to translocate to NMDARs in an activity-dependent manner.” [In: Biology of the NMDA Receptor. Van Dongen, A. M., Ed. Boca Raton (FL), 2009] (see page 2, line 51-56, 59-63). “In particular, the results of the phosphorylation of the NMDA receptor in the process of pain transmission are still controversial. Our previous studies demonstrated that electrical stimulation around acupuncture points suppressed chemotherapy-induced neuropathy via modulation of spinal NR2B phosphorylation [Am J Chin Med, 2015;43(1):57-70], [Brain Res Bull, 2020 Sep;162:237-244], and phosphorylation of spinal NR1 was regulated in animal models of neuropathic pain associated with chronic constrictive injury [J Pain, 2012 Feb;13(2):155-66], [Korean J Physiol Pharmacol, 2012 Dec;16(6):387-92]. In the study of Robert et al., the lumbar spinal NR1 subunit was found to be phosphorylated within 2 hours of the induction of inflammation, and the spinal NR2B expression was rather suppressed by carrageenan-induced inflammation [Mol Pain, 2005 Sep 2;1:25] [39]. Our results demonstrated that carrageenan-induced peripheral inflammatory pain was associated with an increase in phosphorylated NR2B levels in the spinal dorsal horn and glutamate levels in the spinal cord and serum. Thus, the phosphorylation of NMDA receptors is considered a key factor in the development and maintenance of pain caused by exogenous stimuli. Collectively, the findings suggest that glutamate and pNR2B levels in the spinal cord increased after carrageenan injection, and these increases were significantly suppressed by inhibition of TRPV1.” (see page 8, line 267-282).

Minor concerns:

  • For figures 3-5, the authors should indicate that the “vehicle” and “CZP” groups also receive a CR injection. This is implied, but should be explicitly stated in each graph to prevent confusion
    1. Figure 3 groups: CR, Vehicle + CR, CZP + CR
    2. Figure 4 groups: Control, CR, CZP + CR
    3. Figure 5 groups: Control, CR, CZP + CR

Response to Minor comment 1 of the Reviewer #3: 

è Following the reviewer's advice, we revised each group in each graph up to Figure 3-5 to avoid confusion. (see Figure 3, 4 and 5)

  • Abstract line 11: “is an important mediator of various noxious stimuli” is generic. TRPV1 is a mediator of the inflammatory response to various noxious stimuli.

Response to Minor comment 2 of the Reviewer #3: 

è Following the reviewer's comment, we corrected the contents of TRPV1 in the Abstract part as following; “Transient receptor potential vanilloid 1 (TRPV1) has been implicated in peripheral inflammation and is a mediator of the inflammatory response to various noxious stimuli.” (see page 1, line 10-11).

  • Introduction line 56: The sentence “NMDA receptors comprise several amino acid residues at the C-terminal, including calcium/calmodulin-dependent protein kinase II (CaMKII), protein kinase A (PKA), and protein kinase C (PKC)” does not make sense. I think the authors meant to say that NMDA receptors are “regulated by kinases at several amino resides on the C-terminus”

Response to Minor comment 3 of the Reviewer #3: 

è We made a mistake in expressions and references in the writing of the Introduction section. According to the reviewer’s comment, we rechecked the references and further revised the description of the NMDA receptors in the Introduction part as following; “The NMDA receptors composed of 4 subunits derived from the related families of NR1, NR2, and NR3, the typical NMDA receptors consist of two NR1 subunits that bind glycine and two NR2 subunits that bind glutamate [Curr Opin Pharmaco. 2007 Feb;7(1):39-47]. The NR1 is an essential subunit that combines with NR2 or NR3 subunits to form a functional receptor [Neurology. 2011 May 17;76(20):1750-7], and has been implicated in inflammatory pain sensitization during inflammation-induced nociception [Anesthesiology. 2010 Jun;112(6):1482-93]”, “The intracellular domains of NMDA receptor subunits contain consensus phosphorylation sites that regulates NMDA receptors for serine and threonine kinases. The two most actively studied NMDA receptors modulation are protein kinase A (PKA) and protein kinase C (PKC). The calcium/calmodulin-dependent protein kinase II (CAMKII) is also known to translocate to NMDARs in an activity-dependent manner.” [In: Biology of the NMDA Receptor. Van Dongen, A. M., Ed. Boca Raton (FL), 2009] (see page 2, line 52-64).

  • Referring to a group as “normal” seems strange. Better to call them the “control” group.

Response to Minor comment 4 of the Reviewer #3: 

è Based on the reviewers' comments, we replaced the expression 'normal' with the expression 'control' in all parts of the paper.

  • Conclusion line 386: “TRPV1 attenuates inflammatory pain via NMDA receptors” is an overstatement. TRPV1 MAY do this, but the authors provide no evidence.

Response to Minor comment 5 of the Reviewer #3: 

è Based on the reviewers' comments, we removed an overstatement that did not provide evidence in the Conclusion section and revised it as follows: “Collectively, we presume that TRPV1 and NMDA receptors in the spinal cord are implicated in the transmission of inflammatory pain, and inhibition of TRPV1 may attenuates inflammatory pain via NMDA receptors.” (see page 11, line 427-429).

References

  • Horvath G, Kekesi G, Nagy E, and Benedek G. The role of TRPV1 receptors in the antinociceptive effect of anandamide at spinal level. 2008;134(3):277-84.
  • Watanabe M, Ueda T, Shibata Y, Kumamoto N, and Ugawa S. The role of TRPV1 channels in carrageenan-induced mechanical hyperalgesia in mice. 2015;26(3):173-8.
  • Rygh LJ, Svendsen F, Hole K, and Tjolsen A. Increased spinal N-methyl-D-aspartate receptor function after 20 h of carrageenan-induced inflammation. 2001;93(1):15-21.
  • Eisenberg E, LaCross S, and Strassman AM. The effects of the clinically tested NMDA receptor antagonist memantine on carrageenan-induced thermal hyperalgesia in rats. Eur J Pharmacol.1994;255(1-3):123-9.
  • Caudle RM, Perez FM, Del Valle-Pinero AY, and Iadarola MJ. Spinal cord NR1 serine phosphorylation and NR2B subunit suppression following peripheral inflammation. Mol Pain.2005;1:25.

Round 2

Reviewer 1 Report

The revised version of the manuscript has been improved, just the english language at lanes 61-63 needs to be revised.

Reviewer 3 Report

The authors have responded to the majority of my suggestions with regards to data presentation and text edits. However, they provide no additional data. My first major point from the prior review stands. Though the data are sound and experiments well designed, the data presented lack novelty. 

Barring expansion of the study scope, I have no additional issues with the manuscript except the need for editing for English language.